# THE VARIATIONAL HOMOENCODER: LEARNING TO INFER HIGH-CAPACITY GENERATIVE MODELS FROM FEW EXAMPLES

## ABSTRACT

Hierarchical Bayesian methods have the potential to unify many related tasks (e.g. $k$-shot classification, conditional, and unconditional generation) by framing each as inference within a single generative model. We show that existing approaches for learning such models can fail on expressive generative networks such as PixelCNNs, by describing the global distribution with little reliance on latent variables. To address this, we develop a modification of the Variational Autoencoder in which encoded observations are decoded to new elements from the same class; the result, which we call a *Variational Homoencoder* (VHE), may be understood as training a hierarchical latent variable model which better utilises latent variables in these cases. Using this framework enables us to train a hierarchical PixelCNN for the Omniglot dataset, outperforming all existing models on test set likelihood. With a single model we achieve both strong one-shot generation and near human-level classification, competitive with state-of-the-art discriminative classifiers. The VHE objective extends naturally to richer dataset structures such as factorial or hierarchical categories, as we illustrate by training models to separate character content from simple variations in drawing style, and to generalise the style of an alphabet to new characters.

## 1 INTRODUCTION

Learning from few examples is possible only with strong inductive biases. In machine learning such biases can come from hand design, as in the parametrisation of a model, or can be the result of a meta-learning algorithm. Furthermore they may be task-specific, as in discriminative modelling, or may describe the world causally so as to be naturally reused across many tasks.

Recent work has approached one- and few-shot learning from all of these perspectives. Siamese Networks (Koch, 2015), Matching Networks (Vinyals et al., 2016), Prototypical Networks (Snell et al., 2017) and MANNs (Santoro et al., 2016) are all models discriminatively trained for few-shot classification. Such models can achieve state-of-the-art performance at the task they were trained for, but provide no principled means for transferring knowledge to other tasks.

Other work such as Rezende et al. (2016) has developed conditional generative models, which take one or a few observations from a class as input, and return a distribution over new elements $p(x|D)$. These models may be used as classifiers despite not being explicitly trained for this purpose, by comparing conditional likelihoods. They may also be used to generate full sets incrementally as $p(X) = \prod_i p(x_i|x_1, \ldots, x_{i-1})$, as discussed in Generative Matching Networks (Bartunov & Vetrov, 2016). However, such models are a more natural fit to sequences than sets as they typically lack exchangeability, and furthermore they do not expose any latent representation of shared structure within a set.

Finally are hierarchical approaches that model shared structure through latent variables, as $p(X) = \int_c p(c) \prod_i p(x_i|c) \mathrm{d}c$. For example, Lake et al. (2015) develop a compositional causal generative model of handwritten characters, achieving human-level performance at a variety of related tasks. Salakhutdinov et al. (2013) follow a similar Bayesian philosophy while employing a more general-purpose architecture. Most recently the *Neural Statistician* (Edwards & Storkey, 2016) uses amortised variational inference to support learning in a deep hierarchical generative model.

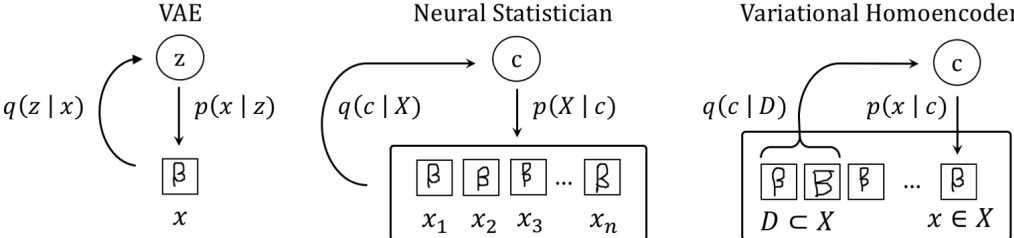

Figure 1: Single step of gradient training in various models. A VAE treats all datapoints as independent, so only a single random element need be encoded and decoded each step. A Neural Statistician instead feeds a full set of elements $X$ through both encoder and decoder networks, in order to share a latent variable $c$. In a VHE, we bound the full likelihood $p(X)$ using only random subsamples $D$ and $x$ for encoding/decoding. Optionally, $p(x|c)$ may be defined through a local latent variable $z$.

In this work we propose the *Variational Homoencoder* (VHE), aiming to combine several advantages of the models described above:

1. Like conditional generative approaches, we train on a few-shot generation objective which matches how our model may be used at test time. However, by introducing an encoding cost, we simultaneously optimise a likelihood lower bound for a hierarchical generative model, in which structure shared across elements is made explicit by shared latent variables.

2. Previous work (Edwards & Storkey, 2016) has learned hierarchical Bayesian models by applying Variational Autoencoders to sets, such as classes of images. However, their approach requires feeding a full set through the model per gradient step (Figure 1), rendering it intractable to train on very large sets. In practice, they avoid computational limits by sampling smaller subset as training data. In a VHE, we instead optimise a likelihood bound for the complete dataset, while *constructing this bound* by subsampling. This approach can not only improve generalisation, but also departs from previous work by extending to models with richer latent structure, for which the joint likelihood cannot be factorised.

3. As with a VAE, the VHE objective includes both an *encoding*- and *reconstruction*- cost. However, by sharing latent variables across a large set of elements, the *encoding cost* per element is reduced significantly. This facilitates use of powerful autoregressive decoders, which otherwise often suffer from ignoring latent variables (Chen et al., 2016). We demonstrate the significance of this by applying a VHE to the Omniglot dataset. Using a Pixel-CNN decoder (Oord et al., 2016), our generative model is arguably the first with a general purpose architecture to both attain near human-level one-shot classification performance and produce high quality samples in one-shot generation.

## 2 BACKGROUND

### 2.1 VARIATIONAL AUTOENCODERS

When dealing with latent variable models of the form $p(x) = \int_z p(z)p(x|z)\mathrm{d}z$, the integration is necessary for both learning and inference but is often intractable to compute in closed form. *Variational Autoencoders* (VAEs, Kingma & Welling (2013)) provide a method for learning such models by utilising neural-network based approximate posterior inference. Specifically, a VAE comprises a generative network $p(z)p(x|z)$ parametrised by $\theta$, alongside a separate inference network $q(z;x)$ parameterised by $\phi$. These are trained jointly to maximise a single objective:

$$\mathcal{L}_X(\theta, \phi) = \sum_{x \in X} \left[ \log p_\theta(x) - \mathrm{D}_{KL}\Big(q_\phi(z;x) \parallel p_\theta(z|x)\Big) \right] \tag{1}$$

$$= \sum_{x \in X} \left[ \mathop{\mathbb{E}}_{q_\phi(z;x)} \log p_\theta(x|z) - \mathrm{D}_{KL}\Big(q_\phi(z;x) \parallel p_\theta(z)\Big) \right] \tag{2}$$

As can be seen from Equation 1, this objective $\mathcal{L}_X$ is a lower bound on the total log likelihood of the dataset $\sum_{x \in X} \log p(x)$, while $q(z; x)$ is trained to approximate the true posterior $p(z|x)$ as accurately as possible. If it could match this distribution exactly then the bound would be tight so that the VAE objective equals the true log likelihood of the data. In practice, the resulting model is typically a compromise between two goals: pulling $p$ towards a distribution that assigns high likelihood to the data, but also towards one which allows accurate inference by $q$. Equation 2 provides a formulation for the same objective which can be optimised stochastically, using Monte-Carlo integration to approximate the expectation.

## 2.2 Variational Autoencoders Over Sets

The *Neural Statistician* (Edwards & Storkey, 2016) is a Variational Autoencoder in which each item to be encoded is itself a set, such as the set $X^{(i)}$ of all images with a particular class label $i$:

$$X^{(i)} = \{x_1^{(i)}, x_2^{(i)}, \cdots, x_n^{(i)}\} \tag{3}$$

The generative model for sets, $p(X)$, is described by introduction of a corresponding latent variable $c$. Given $c$, individual $x \in X$ are conditionally independent:

$$p(X) = \int_c p(c) \prod_{x \in X} p(x|c) \mathrm{d}c \tag{4}$$

This functional form is justified by de Finetti's theorem under the assumption that elements within in each set $X$ are exchangeable. The likelihood is again intractable to compute, but it can be bounded below via:

$$\log p(X) \geq \mathcal{L}_X = \mathop{\mathbb{E}}_{q(c;X)} \left[ \sum_{x \in X} \log p(x|c) \right] - \mathrm{D}_{KL}\big(q(c; X) \parallel p(c)\big) \tag{5}$$

## 3 Variational Homoencoders

Unfortunately, calculating the variational lower bound for each set $X$ requires evaluating both $q(c; X)$ and $p(X|c)$, meaning that the entire set must be passed through both networks for each gradient update. This can easily become intractable for classes with hundreds of examples. Indeed, previous work (Edwards & Storkey, 2016) ensures that sets used for training are always of small size by instead maximising a likelihood lower-bound for randomly sampled subsets.

In this work we instead replace the variational lower-bound in Equation 5 with a new objective, itself constructed via sub-sampled datasets of reduced size. We use a constrained variational distribution $q(c; D), D \subseteq X$ for posterior inference and an unbiased stochastic approximation $\log p(x|c), x \in X$ for the likelihood. In the following section we show that the resulting objective can be interpreted as a lower-bound on the log-likelihood of the data.

This bound will typically be loose due to stochasticity in sampling $D$, and we view this as a regularization strategy: we aim to learn latent representations that are quickly inferable from a small number of instances, and the VHE objective is tailored for this purpose.

### 3.1 Stochastic lower bound

We would like to learn a generative model for sets $X$ of the form

$$p(X) = \int p(c) \prod_{x \in X} p(x|c) \mathrm{d}c \tag{6}$$

We will refer our full dataset as a union of disjoint sets $\mathcal{X} = X_1 \sqcup X_2 \sqcup \ldots \sqcup X_n$, and use $X_{(x)}$ to refer to the set $X_i \ni x$. Using the standard consequent of Jensen's inequality, we can lower bound the log-likelihood of each set $X$ using an arbitrary distribution $q$. In particular, we give $q$ as a fixed function of arbitrary data.

$$\log p(X) \geq \mathop{\mathbb{E}}_{q(c;D)} \log p(X|c) - \mathrm{D}_{KL}\big[q(c; D) \parallel p(c)\big], \quad \forall D \subset X \tag{7}$$

**Algorithm 1:** Minibatch training for the *Variational Homoencoder*. Minibatches are of size $M$. Stochastic inference network $q$ uses subsets of size $N$.

> initialize $(\theta, \phi)$  *Parameters for $p$ and $q$*
> **repeat**
>   sample $(x_k, i_k)$ for $k = 1, \ldots, M$  *Minibatch of elements with corresponding class labels*
>   sample $D_k \subseteq X_{i_k}$ for $k = 1, \ldots, M$  *where $|D_k| = N$*
>   sample $c_k \sim q_\phi(c; D_k)$ for $k = 1, \ldots, M$
>   *(optional)* sample $z_k \sim q_\phi(z; c_k, x_k)$ for $k = 1, \ldots, M$
>   $\mathbf{g} \approx \frac{1}{M} \sum_k \nabla \mathcal{L}_{\theta,\phi}(x_k; D_k, |X_{i_k}|)$  *Reparametrization gradient estimate using $\mathbf{c}, \mathbf{z}$*
>   $(\theta, \phi) \leftarrow (\theta, \phi) + \lambda \mathbf{g}$  *Gradient step, e.g SGD*
> **until** convergence of $(\theta, \phi)$

Splitting up individual likelihoods, we may rewrite

$$\log p(X) \geq \mathop{\mathbb{E}}_{q(c;D)} \Big[ \sum_{x \in X} \log p(x|c) \Big] - \mathrm{D}_{KL}\big[q(c; D) \parallel p(c)\big], \qquad \forall D \subset X \tag{8}$$

$$= \sum_{x \in X} \Big[ \mathop{\mathbb{E}}_{q(c;D)} \log p(x|c) - \frac{1}{|X|} \mathrm{D}_{KL}\big[q(c; D) \parallel p(c)\big] \Big], \qquad \forall D \subset X \tag{9}$$

$$\stackrel{\text{def}}{=} \sum_{x \in X} \mathcal{L}(x; D, |X|), \qquad \forall D \subset X \tag{10}$$

Finally, we can replace the universal quantification with an expectation under any distribution of D (e.g. uniform sampling from X without replacement):

$$\log p(X) \geq \mathop{\mathbb{E}}_{D \subset X} \sum_{x \in X} \mathcal{L}(x; D, |X|) = \sum_{x \in X} \mathop{\mathbb{E}}_{D \subset X} \mathcal{L}(x; D, |X|) \tag{11}$$

$$\log p(\mathcal{X}) \geq \sum_{x \in \mathcal{X}} \mathop{\mathbb{E}}_{D \subset X_{(x)}} \mathcal{L}(x; D, |X_{(x)}|) \tag{12}$$

This formulation suggests a simple modification to the VAE training procedure, as shown in Algorithm 1. At each iteration we select an element $x$, use resampled elements $D \subset X_{(x)}$ to construct the approximate posterior $q(c; D)$, and rescale the encoding cost appropriately. If the generative model $p(x|c)$ also describes a separate latent variable $z$ for each element, we may simply introduce a second inference network $q(z; c, x)$ in order to replace the exact reconstruction error of Equation 9 by a conditional VAE bound:

$$\log p(x|c) \geq \mathop{\mathbb{E}}_{q(z;c,x)} \log p(x|c, z) - \mathrm{D}_{KL}\Big(q(z; c, x) \parallel p(z|c)\Big) \tag{13}$$

## 3.2 APPLICATION TO STRUCTURED DATASETS

The above derivation applies to a dataset partitioned into *disjoint* subsets $\mathcal{X} = X_1 \sqcup X_2 \sqcup \ldots \sqcup X_n$, each with a corresponding latent variable $c_i$. However, many datasets offer a richer organisational structure, such as the hierarchical grouping of characters into alphabets (Lake et al., 2015) or the factorial categorisation of rendered faces by identity, pose and lighting (Kulkarni et al., 2015).

Provided that such organisational structure is known in advance, we may generalise the training objective in Equation 12 to include a separate latent variable $c_i$ for each group $X_i$ within the dataset, even when these groups overlap. To do this we first rewrite this bound in its most general form, where $\mathbf{c}$ collects all latent variables:

$$\log p(\mathcal{X}) \geq \mathop{\mathbb{E}}_{Q(\mathbf{c};\mathbf{D})} \Big[ \sum_{x \in X} \log p(x|\mathbf{c}) \Big] - \mathrm{D}_{KL}\big[Q(\mathbf{c}; \mathbf{D}) \parallel P(\mathbf{c})\big] \tag{14}$$

As shown in Figure 2, a separate $D_i \subset X_i$ may be subsampled for inference of each latent variable $c_i$, so that $Q(\mathbf{c}) = \prod_i q_i(c_i; D_i)$. This leads to an analogous training objective (Equation 15), which

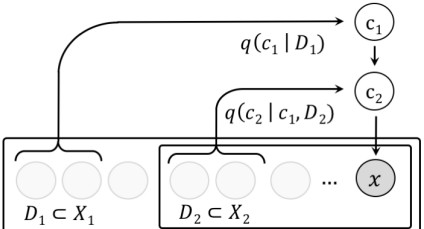 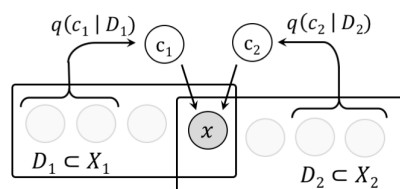

Figure 2: Application of VHE framework to hierarchical (*left*) and factorial (*right*) models. Given an element $x$ such that $x \in X_1$ and $x \in X_2$, an approximate posterior is constructed for the corresponding shared latent variables $c_1, c_2$ using subsampled sets $D_1 \subset X_1, D_2 \subset X_2$

may be applied to data with factorial or hierarchical category structure. For the hierarchical case, this objective may be further modified to infer layers sequentially, as in Supplementary Material 6.2.

$$\log p(\mathcal{X}) \geq \sum_{x \in \mathcal{X}} \mathop{\mathbb{E}}_{\substack{D_i \subset X_i \\ \text{for each} \\ i:x \in X_i}} \left[ \mathop{\mathbb{E}}_{\substack{q_i(c_i;D_i) \\ \text{for each} \\ i:x \in X_i}} \log p(x|\mathbf{c}) - \sum_{i:x \in X_i} \frac{1}{|X_i|} \mathrm{D}_{KL}\big(q_i(c_i;D_i) \parallel p(c_i)\big) \right] \qquad (15)$$

## 3.3 DISCUSSION

### POWERFUL DECODER MODELS

As evident in Equation 9, the VHE objective provides a formal motivation for KL rescaling in the variational objective (a common technique to increase use of latent variables in VAEs) by sharing these variables across many elements. This is of particular importance when using autoregressive decoder models, for which a common failure mode is to learn a decoder $p(x|z)$ with no dependence on the latent space, thus avoiding the encoding cost. In the context of VAEs, this particular issue has been discussed by Chen et al. (2016) who suggest crippling the decoder as a potential remedy.

The same failure mode can occur when training a VAE for sets if the inference network $q$ is not able to reduce its approximation error $\mathrm{D}_{KL}[q(c;D) \parallel p(c|D)]$ to below the total correlation of $D$, either because $|D|$ is too small, or the inference network $q$ is too weak. *Variational Homoencoders* suggest a potential remedy to this, encouraging use of the latent space by reusing the same latent variables across a large set $X$. This allows a VHE to learn useful representations even with $|D| = 1$, while at the same time utilising a powerful decoder model to achieve highly accurate density estimation.

### CONSTRAINED POSTERIOR APPROXIMATION

In a VAE, use of a recognition network encourages learning of generative models whose structure permits accurate amortised inference. In a VHE, this recognition network takes only a small subsample as input, which additionally encourages that the true posterior $p(c|X)$ can be well approximated from only a few examples of $X$. For a subsample $D \subset X$, $q(c;D)$ is implicitly trained to minimise the KL divergence from this posterior *in expectation* over possible sets $X$ consistent with $D$. For a data distribution $p_d$ we may equivalently describe the VHE objective (Equation 12) as

$$\mathop{\mathbb{E}}_{p_d(D)} \mathop{\mathbb{E}}_{p_d(X|D)} \left[ \mathop{\mathbb{E}}_{x \in X} \left[ \log p(x) \right] - \frac{1}{|X|} \mathrm{D}_{KL}\Big[ q(c;D) \parallel p(c|X) \Big] \right] \qquad (16)$$

Note that the variational gap on the right side of this equation is itself bounded by:

$$\mathop{\mathbb{E}}_{p_d(X|D)} \mathrm{D}_{KL}\Big[ q(c;D) \parallel p(c|X) \Big] \geq \mathrm{D}_{KL}\Big[ q(c;D) \parallel \mathop{\mathbb{E}}_{p_d(X|D)} p(c|X) \Big] \geq 0 \qquad (17)$$

The left inequality is tightest when $p(c|X)$ matches $p(c|D)$ well across all $X$ consistent with $D$, and exact only when these are equal. We view this aspect of the VHE loss as regulariser for constrained posterior approximation, encouraging models for which the posterior $p(c|X)$ can be well determined by sampled subsets $D \subset X$. This reflects how we expect the model to be used at test time, and in

practice we have found this 'loose' bound to perform well in our experiments. In principle, the bound may also be tightened by introducing an auxiliary inference network (see Supplementary Material 6.1) which we leave as a direction for future research.

## 4 EXPERIMENTAL RESULTS

### 4.1 SIMPLE 1D DISTRIBUTIONS

With a Neural Statistician model, under-utilisation of latent variables is expected to pose the greatest difficulty either when $|D|$ is too small, or the inference network $q$ is insufficiently expressive. We demonstrate on simple 1D distributions that a Variational Homoencoder can bring improvements under these circumstances. For this we created five datasets as follows, each containing 100 classes from a particular parametric family, and with 100 elements sampled from each class.

1. **Gaussian**: Each class is Gaussian with $\mu$ drawn from a Gaussian hyperprior (fixed $\sigma^2$).
2. **Mixture of Gaussians**: Each class is an even mixture of two Gaussian distributions with location drawn from a Gaussian hyperprior (fixed $\sigma^2$ and separation).
3. **von Mises**: Each class is von Mises with $\mu$ drawn from a Uniform hyperprior (fixed $\kappa$).
4. **Gamma**: Each class is Gamma with fixed $\beta$, and with $\alpha$ drawn from a Uniform hyperprior.
5. **Discrete**: Each class is Uniform on a subset of $\{1,\dots,8\}$, either *1-4*, *5-8*, *odd* or *even*.

For each dataset, we then trained models using a variety of values for $|D|$, restricting the inference network $q(c; D)$ to a simple linear map with Gaussian output. In each case the generative model $p(x|c)$ was set to the correct parametric family, with parameters learned as a linear function of $c$. All models were built in Torch 7 (Collobert et al., 2011) and optimised using Adam (Kingma & Welling, 2013) for 200 epochs. To aid optimisation we used an additional 50 epochs for KL annealing, and used training error to select the best parameters from 3 independent training runs.

Our results show that, when $|D|$ is small, the Neural Statistician often places little to no information in $q(c; D)$ (Figure 3, top row). Our careful training suggests that this is not an optimisation difficulty, but is core to the objective as in Chen et al. (2016). In these cases a VHE better utilises the latent space, leading to improvements in both few-shot generation (by conditional NLL) and classification. Importantly, this is achieved while retaining good likelihood of test-set classes, typically matching or improving upon that achieved by a Neural Statistician (including a standard VAE, corresponding to $|D| = 1$). Please see Supplement 6.3 for further comparison to alternative objectives.

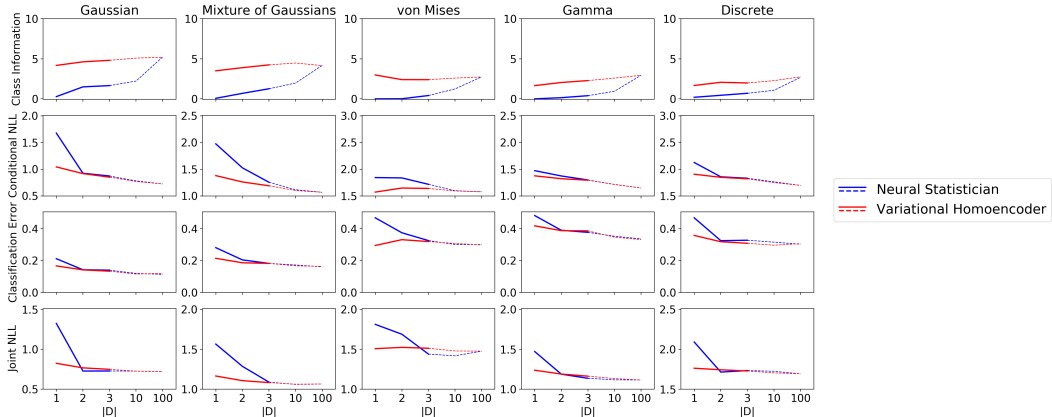

Figure 3: Comparison of models trained using Neural Statistician and VHE objectives. $|D|$ is the number of encoder inputs during training. *Top row*: Mean encoded information $\mathrm{D}_{KL}[q(c; D) \parallel p(c)]$; *Second row*: $|D|$-shot generation loss $-\mathbb{E}_{c \sim q(c;D)} \log p(x'|c)$; *Third row*: $|D|$-shot binary classification error, classified by minimising conditional NLL *Bottom row*: Joint NLL (per element) of full test set, calculated by importance weighting on 200 samples from $q(c; X)$;

### 4.2 HANDWRITTEN CHARACTER CLASSES

To validate our claim that the VHE objective can facilitate learning with more expressive generative networks, we trained a variety of models on the Omniglot dataset to explore the interaction between model architecture and training objective. We consider two model architectures: a standard deconvolutional network based on Edwards & Storkey (2016), and a hierarchical PixelCNN architecture inspired by the recently proposed PixelVAE (Gulrajani et al., 2016). For each, we compare models trained with the Variational Homoencoder objective against three alternative objectives.

For our hierarchical PixelCNN architecture (Figure 4) each character class is associated with a spatial latent variable $c$ (a character 'template') drawn from a learned PixelCNN prior. For each character $x$, a sampled affine transform $t_x$ is applied to this template (Jaderberg et al., 2015) and the result is used to condition a Gated PixelCNN (Oord et al., 2016). For $q(t; x)$ we use a CNN with Batch Normalisation (Ioffe & Szegedy, 2015). For $q(c; D)$ we use a Spatial Transformer followed by a single convolution, and then average over $D$ to output parameters of a diagonal Gaussian distribution.

Using both PixelCNN and deconvolutional architectures, we trained models by several objectives matched in the computational cost per gradient step of training. Firstly, we compare a VHE model against a Neural Statistician baseline, with each trained on sampled subsets $D \subset X$ with $|D| = 5$ (as in Edwards & Storkey (2016)). Secondly, since the VHE introduces both data-resampling and KL-rescaling as modifications to this baseline, we separate the contributions of each using two intermediate objectives:

$$\text{Resample only:} \quad \mathbb{E}_{\substack{D \subset X \\ x \in X}} \left[ \mathbb{E}_{q(c;D)} \log p(x|c) - \frac{1}{|D|} \text{D}_{KL} \big[ q(c; D) \parallel p(c) \big] \right] \tag{18}$$

$$\text{Rescale only:} \quad \mathbb{E}_{\substack{D \subset X \\ x \in D}} \left[ \mathbb{E}_{q(c;D)} \log p(x|c) - \frac{1}{|X|} \text{D}_{KL} \big[ q(c; D) \parallel p(c) \big] \right] \tag{19}$$

All models were trained on a random sample of 1200 Omniglot classes using images scaled to 28x28 pixels, dynamically binarised, and augmented by 8 rotations/reflections to produce new classes. We additionally used 20 small random affine transformations to create new instances within each class. Models were optimised using Adam (Kingma & Welling, 2013), and we used training error to select the best parameters from 5 independent training runs. This was necessary to ensure a fair comparison with the Neural Statistician objective, which otherwise converges to a local optimum with $q(c; D) = p(c)$. We additionally implemented the 'sample dropout' trick of Edwards & Storkey (2016), but found that this did not have any effect on model performance.

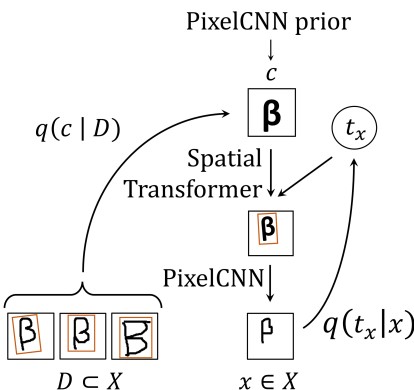

Figure 4: Autoregressive *Variational Homoencoder* for Omniglot characters.

Figure 5: One-shot same-class samples generated by our model. Cues images were drawn at random from previously unseen classes.

Table 1: Interaction of model architecture and training objective on 5-shot, 20 way classification accuracy.

|  | KL / nats* | Accuracy |
|---|---|---|
| **Deconvolutional Architecture** | | |
| NS [7] | 31.34 | 95.6% |
| Resample | 25.74 | 94.0% |
| Rescale | 477.65 | 95.3% |
| *VHE* | 452.47 | 95.6% |
| **PixelCNN Architecture** | | |
| NS | 14.90 | 66.0% |
| Resample | 0.22 | 4.9% |
| Rescale | 506.48 | 62.8% |
| *VHE* | 268.37 | **98.8%** |

*$D_{KL}\big(q(c;D)\parallel p(c)\big)$, train set

Table 2: Comparison of deep learning techniques for Omniglot classification

|  | Accuracy (20-way) | |
|---|---|---|
|  | 1-shot | 5-shot |
| **Generative models**, $\log p(X)$ | | |
| Variational Memory Addressing [2] | 77% | 91 % |
| Generative Matching Networks* [1] | 77.0% | 91.0% |
| Neural Statistician [7] | 93.2% | 98.1% |
| *Variational Homoencoder* | **95.2%** | **98.8%** |
| **Discriminative models**, $\log q(y\mid x, X, Y)$ | | |
| Siamese Networks [16] | 88.1% | 97.0% |
| Matching Networks [26] | 93.8% | 98.7% |
| Convnet with memory module [14] | 95.0% | 98.6% |
| mAP-DLM [25] | 95.4% | 98.6% |
| Model-Agnostic Meta-learning [8] | 95.8% | **98.9%** |
| Prototypical Networks [24] | **96.0%** | **98.9%** |

*Uses train/test split from Lake et al. (2015)

Table 1 collects classification results of models trained using each of the four alternative training objectives, for both architectures. When using a standard deconvolutional architecture, we find little difference in classification performance between all four training objectives, with the Neural Statistician and VHE models achieving equally high accuracy.

For the hierarchical PixelCNN architecture, however, significant differences arise between training objectives. In this case, a Neural Statistician learns an strong global distribution over images but makes only minimal use of latent variables $c$. This means that, despite the use of a higher capacity model, classification accuracy is much poorer (66%) than that achieved using a deconvolutional architecture. For the same reason, conditional samples display an improved sharpness but are no longer identifiable to the cue images on which they were conditioned (Figure 6). Our careful training suggests that this is not an optimisation difficulty but is core to the objective, as discussed in Chen et al. (2016).

By contrast, a VHE is able to gain a large benefit from the hierarchical PixelCNN architecture, with a 3-fold reduction in classification error (5-shot accuracy 98.8%) and conditional samples which are simultaneously sharp and identifiable (Figure 6). This improvement is in part achieved by increased utilisation of the latent space, due to rescaling of the KL divergence term in the objective. However, our results show that this common technique is insufficient when used alone, leading to overfitting to cue images with an equally severe impairment of classification performance (accuracy 62.8%). Rather, we find that KL-rescaling and data resampling must be used together in order to for the benefit of the powerful PixelCNN architecture to be realised.

Table 2 lists the classification accuracy achieved by VHEs with both $|D| = 1$ and $|D| = 5$, as compared to existing deep learning approaches. We find that both networks are not only state-of-the-art amongst deep generative models, but are also competitive against the best discriminative models trained directly for few-shot classification. Unlike these discriminative models, a VHE is also able to generate new images of a character in one-shot, producing samples which are simultaneously realistic and faithful to the class of the cue image (Figure 5).

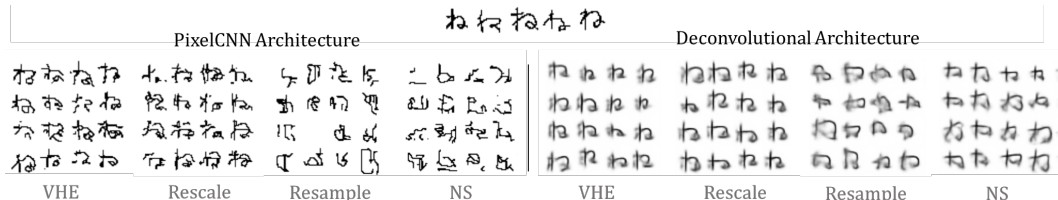

Figure 6: 5-shot samples generated by each model (more in Supplement 6.4.2). With a PixelCNN architecture, both NS and Resample underutilise the latent space and so produce unfaithful samples.

As our goal is to model shared structure across images, we evaluate generative performance using joint log likelihood of the entire Omniglot test set (rather than separately across images). From this perspective, a single element VAE will perform poorly as it treats all datapoints as independent, optimising a sum over log likelihoods for each element. By sharing latent variables across all elements of the same class, a VHE can improve upon this considerably.

Previous work which evaluates likelihood typically uses the train/test split of Burda et al. (2015). However, our most appropriate comparison is with Generative Matching Networks (Bartunov & Vetrov, 2016) as they also model dependencies within a class; thus, we trained models under the same conditions as them, using the harder test split from Lake et al. (2015) with no data augmentation. We evaluate the joint log likelihood of full character classes from the test set, normalised by the number of elements, using importance weighting with k=500 samples from $q(c; X)$. As can be seen in Tables 3 and 4, our hierarchical PixelCNN architecture is able to achieve state-of-the-art log likelihood results only when trained using the full Variational Homoencoder objective.

Table 3: Joint NLL of Omniglot test set, across architectures and objectives

| | Test NLL per image |
|---|---|
| **Deconvolutional Architecture** | |
| NS [7] | 102.84 nats |
| Resample | 110.30 nats |
| Rescale | 109.01 nats |
| *VHE* | 104.67 nats |
| **PixelCNN Architecture** | |
| NS | 73.50 nats |
| Resample | 66.42 nats |
| Rescale | 71.37 nats |
| *VHE* | **61.22 nats** |

Table 4: Comparison of deep generative models by joint NLL of Omniglot test set

| | Test NLL per image |
|---|---|
| **Independent models** | $\frac{1}{n}\log \prod_i p(x_i)$ |
| DRAW [9] | $< 96.5$ nats |
| Conv DRAW [10] | $< 91.0$ nats |
| VLAE [4] | $89.83$ nats |
| **Conditional models** | $\frac{1}{n}\log \prod_i p(x_i\|x_{1:i-1})$ |
| Variational Memory Addressing [2] | $> 73.9$ nats |
| Generative Matching Networks [1] | $62.42$ nats[1] |
| **Shared-latent models** | $\frac{1}{n}\log \mathbb{E}_{p(c)} \prod_i p(x_i\|c)$ |
| *Variational Homoencoder* | **61.22 nats** |

### 4.3 Modelling richer category structure

To demonstrate how the VHE framework may apply to models with richer category structure, we built both a hierarchical and a factorial VHE (Figure 2) using simple modifications to the above architectures. For the hierarchical VHE, we extended the deconvolutional model with an extra latent layer $a$ using the same encoder and decoder architecture as $c$. This was used to encode alphabet level structure for the Omniglot dataset, learning a generative model for alphabets of the form

$$p(\mathcal{A}) = \int p(a) \prod_{X_i \in \mathcal{A}} \int p(c_i|a) \prod_{x_{ij} \in X_i} p(x_{ij}|c_i, a)\mathrm{d}c_i\mathrm{d}a \tag{20}$$

---

[1]We thank the authors of Bartunov & Vetrov (2016) for providing us with this comparison.

Again, we trained this model using a single objective, using separately resampled subsets $D^a$ and $D^c$ to infer each latent variable (Supplement 6.2). We then tested our model at both one-shot character generation and 5-shot alphabet generation, using samples from previously unseen alphabets. As shown in Figure 7, our single trained model is able to learn structure at both layers of abstraction.

For the factorial VHE, we extended the Omniglot dataset by assigning each image to one of 30 randomly generated styles (independent of its character class), modifying both the colour and pen stroke of each image. We then extended the PixelCNN model to include a 6-dimensional latent variable $s$ to represent the *style* of an image, alongside the existing $c$ to represent the *character*. We used a CNN for style encoder $q(s|D^s)$, and for each image location we condition the PixelCNN decoder using the outer product $s \otimes c_{ij}$.

We then test this model on a *style transfer* task by feeding separate images into the character encoder $q(c|D^c)$ and style encoder $q(s|D^s)$, then rendering a new image from the inferred $(c, s)$ pair. We find that synthesised samples are faithful to the respective character and style of both cue images (Figure 8), demonstrating ability of a factorial VHE to successfully disentangle these two image factors using separate latent variables.

Figure 7: Conditional samples from both character (top) and alphabet (bottom) levels of the same hierarchical model.

Figure 8: Previously unseen characters redrawn in a style inferred from another image. Top two images denote the content (left) and style (right).

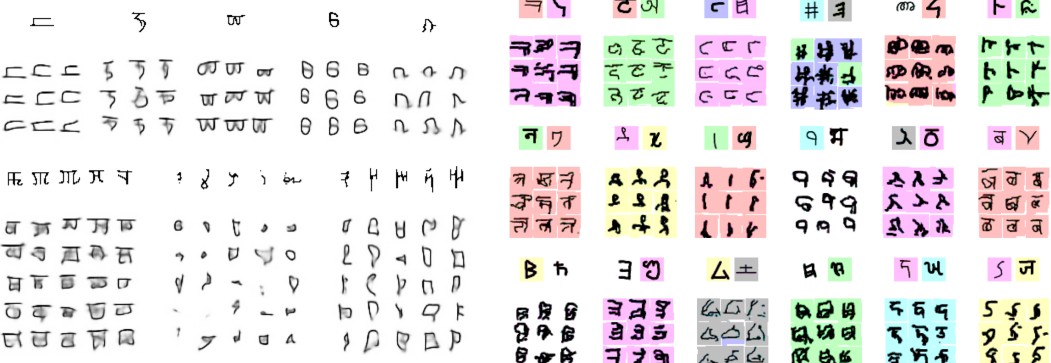

## 5 CONCLUSION AND FUTURE WORK

We introduced the *Variational Homoencoder*: a deep hierarchical generative model learned by a novel training procedure which resembles few-shot generation. This framework allows latent variables to be shared across a large number of elements in a dataset, encouraging them to be well utilised even alongside highly expressive decoder networks. We demonstrate this by training a hierarchical PixelCNN model on Omniglot dataset, and show that our novel training objective is responsible for the state-of-the-art results it achieves. This model is arguably the first which uses a general purpose architecture to both attain near human-level one-shot classification performance and produce high quality samples in one-shot generation.

The VHE framework extends naturally to models with richer latent structure, as we demonstrate with two examples: a hierarchical model which generalises the style of an alphabet to produce new characters, and a factorial model which separates the content and drawing style of coloured character images. In addition to these modelling extensions, our variational bound may also be tightened by learning a subsampling procedure $q(D; X)$, or by introducing an auxiliary inference network $r(D; c, X)$ as discussed in Supplementary Material 6.1. While such modifications were unnecessary for our experiments on Omniglot character classes, we expect that they may yield improvements on other datasets with greater intra-class diversity.

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

## 6 SUPPLEMENTARY MATERIAL

### 6.1 TIGHTENED VARIATIONAL BOUND

The likelihood lower bound in the VHE objective may also be tightened by introduction of an auxiliary network $r(D; c, X)$, trained to infer which subset $D \subset X$ was used in $q$. This meta-inference approach was introduced in Salimans et al. (2015) to develop stochastic variational posteriors using MCMC inference, and has recently been applied to approximate inference evaluation (Cusumano-Towner & Mansinghka, 2017). Applied to Equation 12, this yields a modified bound for the VHE objective

$$\log p(\mathcal{X}) \geq \sum_{x \in \mathcal{X}} \mathop{\mathbb{E}}_{\substack{q'(D; X_{(x)}) \\ q(c; D)}} \left[ \log p(x|c) - \frac{1}{|X_{(x)}|} \log \frac{p(c) r(D; c, X_{(x)})}{q'(D; X_{(x)}) q(c; D)} \right] \tag{21}$$

where $q'(D; X)$ describes the stochastic sampling procedure for sampling $D \subset X$, which indeed may itself be learned using policy gradients.

We have conducted preliminary experiments using fixed $q'$ and a simple functional form $r(D; c, X) = \prod_i r(d_i; c, X) \propto \prod_i \left[ f_\psi(c) \cdot \xi_{d_i} \right]$, learning parameters $\psi$ and embeddings $\{\xi_d : d \in \mathcal{X}\}$; however, on the Omniglot dataset we found no additional benefit over the strictly loose bound (Equation 12). We attribute this to the already high similarity between elements of the same Omniglot character class, allowing the approximate posterior $q(c; D)$ to be relatively robust to different choices of $D$. However, we expect that the gain from using such a tightened objective may be much greater for domains with lower intra-class similarity (e.g. natural images), and thus suggest the tightened bound of Equation 21 as a direction for future research.

### 6.2 VARIATIONAL BOUND FOR HIERARCHICAL MODELS

The resampling trick may be applied iteratively, to construct likelihood bounds over hierarchically organised data. Expanding on Equation 12, suppose that we have collection of datasets

$$\mathbf{X} = \mathcal{X}_1 \sqcup \mathcal{X}_2 \sqcup \ldots \sqcup \mathcal{X}_N \tag{22}$$

For example, each $\mathcal{X}$ might be a different alphabet whose latent description $a$ generates many character classes $X_i$, and for each of these a corresponding latent $c_i$ is used to generate many images $x_{ij}$. From this perspective, we would like to learn a generative model for alphabets $\mathcal{X}$ of the form

$$p(\mathcal{X}) = \int p(a) \prod_{X_i \subset \mathcal{X}} \int p(c|a) \prod_{x \in X_i} p(x|c, a) \mathrm{d}c \mathrm{d}a \tag{23}$$

Reapplying the same trick as before yields a bound taken over all elements $x$:

$$\log p(\mathbf{X}) \geq \sum_{x \in \mathbf{X}} \mathop{\mathbb{E}}_{\substack{D^a \subset \mathcal{X}_{(x)} \\ D^c \subset X_{(x)}}} \left[ \mathop{\mathbb{E}}_{\substack{q_a(a|D_1) \\ q_c(c|D_2, a)}} \log p(x|c) - \frac{1}{|\mathcal{X}_{(x)}|} \mathrm{D}_{KL}\big(q_a(a|D^a) \parallel p(a)\big) \right.$$
$$\left. - \frac{1}{|X_{(x)}|} \mathrm{D}_{KL}\big(q_c(c|D^c, a) \parallel p(c|a)\big) \right] \tag{24}$$

This suggests an analogous *hierachical resampling* procedure: Summing over every element $x$, we can bound the log likelihood of the full hierarchy by resampling subsets $D^c$, $D^a$, etc. at each level to construct an approximate posterior. All networks are trained together by this single objective, sampling $x$, $D^a$ and $D_c$ for each gradient step. Note that this procedure need only require passing sampled elements, rather than full classes, into the upper-level encoder $q_a$.

## 6.3 ADDITIONAL RESULTS ON SIMPLE 1D DISTRIBUTIONS

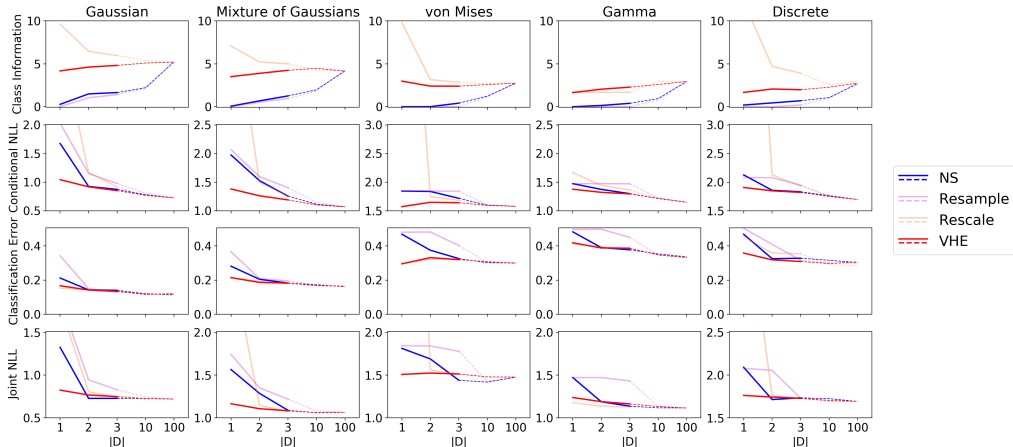

Figure 9: Comparison of models trained simple 1D distributions using various alternate objectives. $|D|$ is the number of encoder inputs during training. *Top row*: Mean encoded information $D_{KL}[q(c; D) \parallel p(c)]$; *Second row*: $|D|$-shot generation loss $-\mathbb{E}_{c \sim q(c;D)} \log p(x'|c)$; *Third row*: $|D|$-shot binary classification error, by minimising conditional NLL *Bottom row*: Joint NLL (per element) of full test set, calculated by importance weighting on 200 samples from $q(c; X)$;

## 6.4 PIXELCNN OMNIGLOT ARCHITECTURE

### 6.4.1 METHODOLOGY

Our architecture uses a 8x28x28 latent variable $c$, with a full architecture detailed below. For our classification experiments, we trained 5 models on each of the objectives (VHE, Rescale only, Resample only, NS). Occasionally we found instability in optimisation, causing sudden large increases in the training objective. When this happened, we halted and restarted training. All models were trained for 100 epochs on 1000 characters from the training set (the remaining 200 have been used as validation data for model selection). Finally, for each objective we selected the parameters achieving the best training error.

Note that we did not optimise or select models based on classification performance, other than through our development of our model's architecture. However, we find that classification performance is well correlated the generative training objective, as can be seen in the full table of results.

We perform classification by calculating the expected conditional likelihood under the variational posterior: $\mathbb{E}_{q(c;D)} p(x|c)$. This is approximated using 20 samples for the outer expectation, and importance sampling with $k = 10$ for the inner integral $p(x|c) = \mathbb{E}_{q(t|x)} \frac{p(t)}{q(t|x)} p(x|c, t)$

To evaluate and compare log likelihood, we trained 5 more models with the same architecture, this time on the canonical 30-20 alphabet split of Lake et al. We did not augment our training data. Again, we split the background set into training data (25 alphabets) and validation data (5) but do not use the validation set in training or evaluation for our final results. We estimate the total class log likelihood by importance weighting, using k=20 importance samples of the class latent $c$ and k=10 importance samples of the transformation latent $t$ for each instance.

6.4.2 CONDITIONAL SAMPLES ON OMNIGLOT

**Baseline Architecture**

Neural Statistician [10]

Resample Only

Rescale Only

Variational Homoencoder

**PixelCNN Architecture**

Neural Statistician

Resample Only

Rescale Only

Variational Homoencoder

### 6.4.3 Model Specification

[d] denotes a dimension d tensor. {t} denotes a set with elements of type t. Posteriors $q$ are Gaussian.

P(C)

A PixelCNN with autoregressive weights along only the spatial (not depth) dimensions of c. We use 2 layers of masked 64x3x3 convolutions, followed by a ReLU and two 8x1x1 convolutions corresponding to the mean and log variance of a Gaussian posterior for the following pixel.

P(T)

t: [16]   Normal(0, 1)

P(X—C,T)

c: [8x28x28], t: [16] $\mapsto$ x: [1x28x28]

| Input | Operation | Output |
|---|---|---|
| t | Linear | t2: [6] |
| c,t2 | Spatial Transformer | y1: [8x28x28] |
| y1 | 64x3x3 Conv; Relu; BatchNorm | y2: [64x28x28] |
| y2 | 64x3x3 Conv; Relu; BatchNorm | y3: [64x28x28] |
| y3 | 64x3x3 Conv; Relu; BatchNorm | y4: [64x28x28] |
| y4 | 64x3x3 Conv; Relu; BatchNorm | y5: [64x28x28] |
| y5 | 64x3x3 Conv; Relu; BatchNorm | y: [64x28x28] |
| y | PixelCNN | x: [1x28x28] |

PixelCNN is gated by y, and is autoregressive along only the spatial (not depth) dimensions of c. We use 2 layers of masked 64x3x3 convolutions, followed by a ReLU, a 2x1x1 convolution and a softmax, corresponding to a Bernoulli distribution on the following pixel.

Q(C;D)

D: {[1x28x28]} $\mapsto$ c: [8x28x28]

| Input | Operation | Output |
|---|---|---|
| D | STNq | Y: {[1x28x28]} |
| Y | Mean | y: [1x28x28] |
| y | 16x28x28 Conv | mu: [8x28x28], logvar: [8x28x28] |

Q(T;X)

x: [1x28x28] $\mapsto$ t: [16]

| Input | Operation | Output |
|---|---|---|
| x | 32x3x3 Conv; 2x2 Max Pooling; ReLU; BatchNorm | y1: [32x15x15] |
| y1 | 32x3x3 Conv; 2x2 Max Pooling; ReLU; BatchNorm | y2: [32x8x8] |
| y2 | 32x3x3 Conv; 2x2 Max Pooling; ReLU; BatchNorm | y3: [32x4x4] |
| y3 | 32x3x3 Conv; 2x2 Max Pooling; ReLU; BatchNorm | y4: [32x2x2] |
| y4 | 32x3x3 Conv; 2x2 Max Pooling; ReLU; BatchNorm | y5: [32x1x1] |
| y5 | Linear | mu: [16], logvar: [16] |

SPATIAL TRANSFORMER STNQ

x: [1x28x28] $\mapsto$ y: [1x28x28]

| Input | Operation | Output |
|-------|-----------|--------|
| x | 16x3x3 Conv; 2x2 Max Pooling; ReLU; BatchNorm | y1: [16x15x15] |
| y1 | 16x3x3 Conv; 2x2 Max Pooling; ReLU; BatchNorm | y2: [16x8x8] |
| y2 | 16x3x3 Conv; 2x2 Max Pooling; ReLU; BatchNorm | y3: [16x4x4] |
| y3 | 16x3x3 Conv; 2x2 Max Pooling; ReLU; BatchNorm | y4: [16x2x2] |
| y4 | 16x3x3 Conv; 2x2 Max Pooling; ReLU; BatchNorm | y5: [16x1x1] |
| y5 | Linear | y6: [6] |
| x, y6 | Spatial Transformer Network | y: [1x28x28] |

## 6.5 HIERARCHICAL OMNIGLOT ARCHITECTURE

We extend the same architecture described in Appendix B of (Edwards & Storkey, 2016), with only a simple modification: we introduce a new latent layer containing a 64-dimensional variable $a$, with a Gaussian prior. We give $p(c|a)$ the same functional form as $p(z|c)$, and give $q(a|D^a)$ the same functional form as $q(c; D^c)$ using the shared encoder.

Figure 10: 10-shot alphabet generation samples from the hierarchical model.

## 6.6 CONDITIONAL SAMPLES ON SILHOUETTES DATASET

We created a VHE using the same deconvolutional architecture as applied to omniglot, and trained it on the Caltech-101 Silhouettes dataset. 10 object classes were held out as test data, which we use to generate both 1-shot and 5-shot conditional samples.

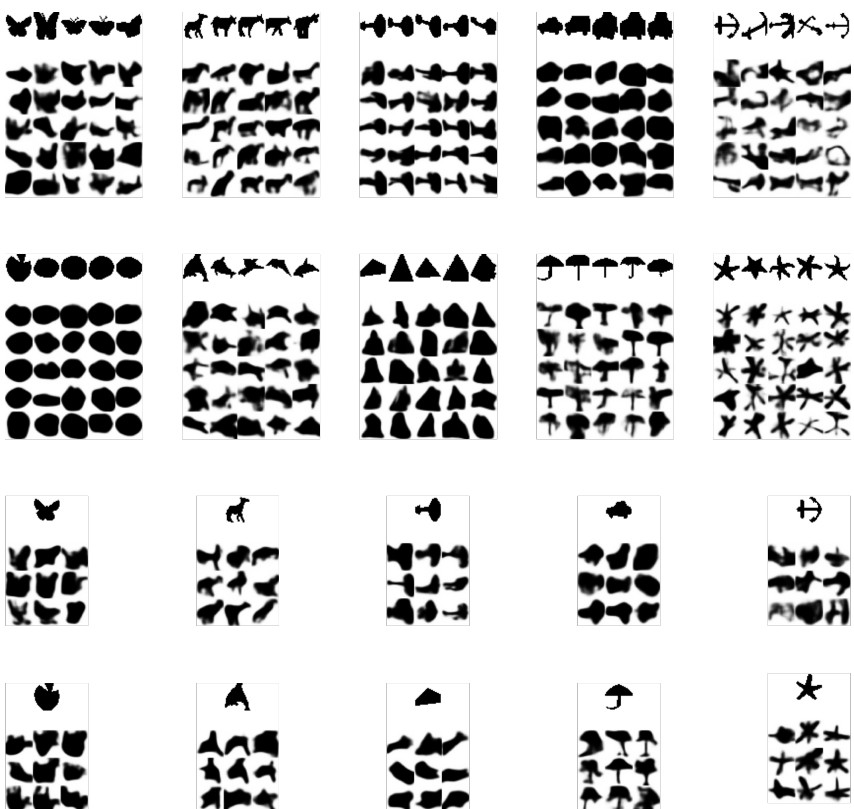

