# OpenReview forum: "The Variational Homoencoder: Learning to Infer High-Capacity Generative Models from Few Examples"
_ICLR.cc/2018/Conference — Reject_

### Official Review · AnonReviewer2 · 2017-11-27
**Interesting, though still avoiding the toughest questions**

**Rating:** 7
**Confidence:** 4

**Review:**

This paper presents an alternative approach to constructing variational lower bounds on data log likelihood in deep, directed generative models with latent variables. Specifically, the authors propose using approximate posteriors shared across groups of examples, rather than posteriors which treat examples independently. The group-wise posteriors allow amortization of the information cost KL(group posterior || prior) across all examples in the group, which the authors liken to the "KL annealing" tricks that are sometimes used to avoid posterior collapse when training models with strong decoders p(x|z) using current techniques for approximate variational inference in deep nets.

The presentation of the core idea is solid, though it did take two read-throughs before the equations really clicked for me. I think the paper could be improved by spending more time on a detailed description of the model for the Omniglot experiments (as illustrated in Figure 3). E.g., explicitly describing how group-wise and per-example posteriors are composed in this model, using Equations and pseudo-code for the main training loop, would have saved me some time. For readers less familiar with amortized variational inference in deep nets, the benefit would be larger.

I appreciate that the authors developed extensions of the core method to more complex group structures, though I didn't find the related experiments particularly convincing.

Overall, I like this paper and think the underlying group-wise posterior construction trick is worth exploring further. Of course, the elephant in the room is how to determine the groups across which the posteriors can be shared and their information costs amortized.

---

> ### Author Response · Authors · 2018-01-05
> **Improvements to clarify & results**
>
> > It did take two read-throughs before the equations really clicked for me
> To improve the clarity of our training procedure, particularly for readers less familiar with amortized variational inference, we have now added pseudocode for the main training loop. We have also elaborated on section 3.1 to explicitly detail how a per-element latent variable z may be added to the bound. Thanks for the suggestions!
>
> > I appreciate that the authors developed extensions of the core method to more complex group structures, though I didn't find the related experiments particularly convincing.
> We have since slightly revised the architecture used in our style/content factorisation experiments, leading to a more powerful model which is able to adapt both colour and pen stroke simultaneously. We hope that the reviewer finds our revised factorial model at least somewhat more convincing.
>
> > Of course, the elephant in the room is how to determine the groups across which the posteriors can be shared and their information costs amortized.
> Indeed, in this paper we tackle only the problem of learning with known labels, but discovering the structure of a dataset unsupervised is an interesting problem and potential future direction! We’d be particularly keen to see our approach embedded within a larger EM-like training algorithm, alternating gradient steps with reassignment of elements to groups. From a good initialisation, we expect that this approach could go a long way towards learning such expressive mixture models from only unlabelled data.

---

> > ### Comment · AnonReviewer2 · 2018-01-12
> > **writing improvements**
> >
> > Thank you for making edits to the technical portion of the paper. I believe the changes improve the paper's readability.

---

### Official Review · AnonReviewer3 · 2017-11-28
**An interesting work on having VAEs for few-shot generation. Some tuning is need though, I think.**

**Rating:** 5
**Confidence:** 5

**Review:**

- Good work on developing VAEs for few-shot learning.
- Most of the results are qualitative and I reckon the paper was written in haste.
- The rest of the comments are below:

- 3.1: I got a bit confused over what X actually is:
 -- "We would like to learn a generative model for **sets X** of the form".
 --"... to refer to the **class X_i** ...".
 -- "we can lower bound the log-likelihood of each **dataset X** ..."

- 3.2: "In general, if we wish to learn a model for X in which each latent variable ci affects some arbitrary subset Xi of the data (**where the Xi may overlap**), ...": Which is just like learning a Z for a labeled X but learning it in an unsupervised manner, i.e. the normal VAE, isn't it? If not, could you please elaborate on what is different (in the case of 3.2 only, I mean)? i.e. Could you please elaborate on what's different (in terms of learning) between 3.2 and a normal latent Z that is definitely allowed to affect different classes of the data without knowing the classes?

- Figure 1 is helpful to clarify the main idea of a VHE.

- "In a VHE, this recognition network takes only small subsets of a class as input, which additionally ...": And that also clearly leads to loss of information that could have been used in learning. So there is a possibility for potential regularization but there is definitely a big loss in estimation power. This is obviously possible with any regularization technique, but I think it is more of an issue here since parts of the data are not even used in learning.

- "Table 4.1 compares these log likelihoods, with VHE achieving state-of-the-art. To": Where is Table 4.1??

- This is a minor point and did not have any impact on the evaluation but VAE --> VHE, reparameterization trick --> resampling trick. Maybe providing rather original headings is better? It's a style issue that is up to tastes anyway so, again, it is minor.

- "However, sharing latent variables across an entire class reduces the encoding cost per element is significantly": typo.

- "Figure ?? illustrates".

---

> ### Author Response · Authors · 2018-01-05
> **Quantitative Results and Clarifications**
>
> > “I reckon the paper was written in haste” “I got a bit confused over what X actually is”
> We greatly apologise that our initial submission contained typos and broken references. These have been fixed in our revised submission, and much of the language has also been updated to improve consistency. We now use ‘set’ for all elements which share a particular latent variable (e.g. all images of a particular character) and ‘dataset’ for the collection of all such sets (e.g. “the omniglot dataset”). Thanks for highlighting this point of unclarity - we hope that our modification will make the exposition clearer for future readers.
>
> > Most of the results are qualitative
> For our revised submission, we include a new section (4.1) in the main paper which compares VHE and Neural Statistician objectives on five simple synthetic datasets, aiming to provide stronger empirical support for the theoretical motivations of section 3. In Omniglot experiments, we now also include quantitative results comparing the generative performance for all eight architecture/objective combinations (Table 3), in addition to the classification accuracy results provided in Table 1.
>
> > Could you please elaborate on what's different (in terms of learning) between 3.2 and a normal latent Z that is definitely allowed to affect different classes of the data without knowing the classes?
> The extended VHE objective allows us to learn latent variable models for structured datasets when the structure is known in advance. For example, our factorial objective might be applied to a dataset of rendered 3D faces when each image is labelled by identity, pose and lighting conditions (as in Deep Convolutional Inverse Graphics Network, Kulkarni et al. 2015) by introducing a separate latent variable for each identity, each pose and each lighting condition.
> We do not tackle unsupervised learning of such categorical structure in this paper, although we’d be keen to see our approach embedded within a larger EM-like training algorithm, alternating gradient steps with reassignment of elements to groups.
>
> > There is a possibility for potential regularization but there is definitely a big loss in estimation power.
> This loss in estimation power is a significant limitation of our method; indeed, it was a great surprise to us that our model achieved such strong results despite this! We attribute its success to the high similarity between Omniglot images within the same class, allowing the approximate posterior q(c; D) to remain relatively robust to different choices of D. However, we expect that reduced estimation power may pose a greater challenge in domains with greater intra-class variation (such natural images) and with this in mind propose an alternative objective in Supplement 6.1 which may tighten the variational bound using an auxiliary inference network. Experimentation in this setting remains a direction of future research.
> We’d also like to note the tricky comparison to existing work with regard to estimation power. The Neural Statistician employs a tighter variational bound, but does so for only an approximate marginal likelihood (based on subsampled training sets). This approximation itself carries unbounded error with respect to the true likelihood. By contrast, the VHE objective uses resampling only within the inference network, trading off estimation power in order to provide a true lower bound on the likelihood of the complete training set.

---

### Official Review · AnonReviewer1 · 2017-11-29
**Conceptually only incremental; but generally good paper.**

**Rating:** 6
**Confidence:** 3

**Review:**

The paper presents some conceptually incremental improvements over the models in “Neural Statistician” and “Generative matching networks”. Nevertheless, it is well written and I think it is solid work with reasonable convincing experiments and good results. Although, the authors use powerful PixelCNN priors and decoders and they do not really disentangle to what degree their good results rely on the capabilities of these autoregressive components.

---

> ### Author Response · Authors · 2018-01-05
> **Disentangling the necessity of our hierarchical PixelCNN architecture**
>
> > Although, the authors use powerful PixelCNN priors and decoders and they do not really disentangle to what degree their good results rely on the capabilities of these autoregressive components
>
> In our revised submission, we compare our hierarchical PixelCNN against a standard deconvolutional baseline model on both generation and classification. In these experiments we find that the expressive PixelCNN architecture can improve results significantly, but that our novel training objective is necessary for gaining this improvement. In particular, amongst the four alternative training objectives we tested, only the VHE was able to utilise the more expressive architecture without suffering from either overfitting or latent degeneracy (Tables 1 & 3, Figure 6). We have now modified the title and abstract of the paper to more strongly emphasise this relationship between training objective and model architecture.

---

### Author Response · Authors · 2018-01-05
**Revised Submission**

We are most grateful for the time and thoughtful comments offered by our reviewers, and delighted by the generally positive sentiment towards the ideas present in this paper. We have posted a revised version in which we aim to address each reviewer’s specific concerns (individual comments are given below). The main changes are the following:
- Rewriting the abstract/title to emphasise the relationship between our VHE objective and the PixelCNN architecture we apply it to.
- Elaborating the training procedure (Algorithm 1)
- Moving the synthetic data experiments to the main paper (Section 4.1)
- Adding quantitative evaluation of generative performance on all 8 architecture/objective variants (Table 3)
- Minor architectural modification on factorial architecture (Section 4.3), to obtain improved results (Figure 8)
- Moving Silhouette generation experiments to Supplementary Material for space.

We believe that addressing these comments has significantly improved the presentation of our work, and hope that this improvement justifies the increased length of our paper (now 10 pages).

---

### Decision · Program_Chairs · 2018-01-29
**ICLR 2018 Conference Acceptance Decision**

**Decision:**

Reject

**Comment:**

Thank you for submitting your paper to ICLR. The reviewers agree that the idea of sharing the approximating distribution across sets of variables is an interesting one and that the Omniglot experiments are thorough. However, although the authors make the nice addition of some simple examples during the revision period and a new table of quantitative results on Omniglot, the consensus is that the experimental results are not quite persuasive enough for publication. Adding a second dataset, such as mini-imagenet or the youtube faces dataset, would make the paper very strong.